# Large-Scale Flow in Micro Electrokinetic Turbulent Mixer

**DOI:** 10.3390/mi11090813

**Published:** 2020-08-28

**Authors:** Keyi Nan, Zhongyan Hu, Wei Zhao, Kaige Wang, Jintao Bai, Guiren Wang

**Affiliations:** 1State Key Laboratory of Photoelectric Technology and Functional Materials, International Scientific and Technological Cooperation Base of Photoelectric Technology and Functional Materials and Application, Institute of Photonics and Photon-technology, Northwest University, Xi’an 710069, China; nancynky@stumail.nwu.edu.cn (K.N.); zyhu@stumail.nwu.edu.cn (Z.H.); wangkg@nwu.edu.cn (K.W.); baijt@nwu.edu.cn (J.B.); 2Department of Mechanical Engineering, University of South Carolina, Columbia, SC 29208, USA; 3Biomedical Engineering Program, University of South Carolina, Columbia, SC 29208, USA

**Keywords:** micro electrokinetic (μEK) turbulence, turbulent mixer, three-dimensional mean flow field, unbalanced electroosmotic flow

## Abstract

In the present work, we studied the three-dimensional (3D) mean flow field in a micro electrokinetic (μEK) turbulence based micromixer by micro particle imaging velocimetry (μPIV) with stereoscopic method. A large-scale solenoid-type 3D mean flow field has been observed. The extraordinarily fast mixing process of the μEK turbulent mixer can be primarily attributed to two steps. First, under the strong velocity fluctuations generated by μEK mechanism, the two fluids with different conductivity are highly mixed near the entrance, primarily at the low electric conductivity sides and bias to the bottom wall. Then, the well-mixed fluid in the local region convects to the rest regions of the micromixer by the large-scale solenoid-type 3D mean flow. The mechanism of the large-scale 3D mean flow could be attributed to the unbalanced electroosmotic flows (EOFs) due to the high and low electric conductivity on both the bottom and top surface.

## 1. Introduction

Microfluidics has many advantages to manipulate fluid flows and target samples in a small size and has broad applications in biomedicine, organic synthesis, electrical engineering, and chemical engineering etc. [1,2,3,4,5] For instance, various micro sensors [6], implantable chips [7], DNA sequencing chips [8], and micromixers [9] etc. have been developed. Micromixer is a core part in integrated microfluidic systems and has been rapidly developed in recent years. It is mainly used to achieve rapid and thorough mixing of multiple samples in microdevices. Due to the advantages of portability, rapid and controllable mixing and integrability [10,11,12], micromixer has been widely used in biochip [13], chemical reactor [14], drug mixing [15], environment [16] and energy [17] etc.

In micromixers, the bulk flow Reynolds numbers ($Re=\rho_{f}Ud/\mu$) is typically far below 2300, the flow is laminar and stable. Here, U is the velocity bulk flow, $\rho_{f}$ is the density of fluid, μ is the dynamic viscosity of fluid and d is the characteristic transverse scale. The mixing is mainly achieved by molecular diffusion and laminar dispersion. Therefore, rapid mixing of different fluids can be hardly realized in micromixer and the mixing efficiency is normally low [18,19]. However, fast mixing and high mixing efficiency are crucial in many applications, e.g., reduce by-products of chemical reactions during mixing [20,21], studying protein folding kinetics [22], genetic diagnosis [23], genetic testing [24], cell sorting [25] and etc.

In order to improve mixing efficiency of micromixers, various mechanisms have been applied. Generally speaking, micromixers can be divided into two categories, which are passive and active micromixers.

Passive micromixer is a kind of micromixers which achieve mixing enhancement with the specially designed shape or structures of microchannel. It includes lamination [26,27], injection [28], chaotic advection [29], and droplet micromixer [30] etc. In passive micromixer, no additional energy is required to generated flow disturbance, therefore, passive micromixer has relatively low energy consumption and low cost accordingly [31,32]. Since the flow is laminar, diffusion and laminar dispersion are the only effective way for fluid mixing. Considering the diffusion time scale ($\tau \sim l^{2}/D$) is proportional to mixing length $l^{2}$, to enhance mixing, the passive mixer generally adopts complex channel structures to increase the contact area by stretching and folding, and reduce the diffusion time by decreasing l between the two fluids. In 2002, Stroock et al. [18] designed a staggered herringbone mixer (SHM). A series of rotational local flows can be generated by changing the shape of the grooves, and a chaotic flow can be thus generated to enhance mixing in the range of 0<Re<100. When Péclet number (Pe) which represents a ratio between convection and diffusion reaches $2\times 10^{3}$, the mixing can reach 90% at a mixing length of 0.7 cm. Later, Hong et al. [33] designed a passive micromixer with modified Tesla structures to produce a Poiseuille flow in the transverse direction of the flow. Both the experimental (nearly 90% mixing) and simulation results (nearly 95% mixing within a length of 7 mm) show that fast mixing can be achieved over a wide range of flow rates. In 2012, Li et al. [34] designed a planar labyrinth mixer which has “S-shaped” structures. They found this mixer could improve mixing efficiency and realizes nearly full mixing in 10 cm mixing distance. The mixing times of achieving 80% mixing are 9.8 and 0.032 s at Re = 2.5 and 30 respectively. This means fast and uniform mixing can be achieved at intermediately low Reynolds numbers (2.5 < Re < 30). Hossain et al. [35] designed a two-layer serpentine crossing microchannels in 2017. Both the numerical and experimental analyses confirm that at least 96% mixing in the micromixer can be achieved throughout the whole Reynolds number range (Re = 0.2–120). At low Reynolds numbers where Re= 0.2–10, the micromixer showed about 99% mixing at the exit. In 2017, Izadi et al. [36] fabricated a mixer to study protein folding. By shrinking the width of microchannel of the order of 1 µm, and applying filter array to further reduce mixing scale l, they claimed that the mixing time can be reduced to astonishingly 8 μs, if the high flow resistance at 1–10 m/s flow speed is not a concern. However, for most of the passive mixers, they still suffer from long mixing time and long mixing length.

Active micromixer is the other kind of micromixer realized by applying externally physical excitations to generate flow disturbance and enhance mixing. The active micromixers include, for instance, acoustic disturbance [37], dielectrophoretic disturbance [38], electrokinetic (EK) flow [39,40,41], and many others. Compared with the passive micromixers, the active micromixers have the advantages of faster mixing with less mixing time and mixing length within a small equipment volume.

EK flow has been widely used to manipulate flows and drive fluids through electrostatic force [42,43] as active micromixers. In 1998, Ramos et al. [44] reviewed the possible mechanisms of driving microflow and enhancing transportation of species by AC electrokinetics, including electrothermal, electroosmotic flow, dielectrophoresis and etc. Around 2000, Ramos and his colleagues [45,46,47] made a series of investigations on fluid transport and flow manipulation by AC EK flows. They found that the fluid flow velocity is related to the polarization of the electrode under the action of non-uniform electric field, and derived the expression of AC electroosmosis velocity. Then, from the experimental and theoretical aspects, it is verified that the fluid flow on the electrode surface is driven by AC electroosmosis, and the variation law of the volume flow driven by the surface velocity with frequency is obtained. Later, Wu et al. [39] designed an active mixer with asymmetric herringbone electrodes and periodic voltage control. The mixing efficiency of this mixer can exceed 90% at a distance of 5 mm downstream of the T-junction. Then, Park et al. [40] utilized electrokinetic instability (EKI) with downstream herringbone-shape structures in a micro-T channel and realized 65% mixing at 2.25 mm from channel inlet. Chen et al. [48] and Posner and Santiago [41] initiated systematical studies on EKI. Then, Posner et al. [49] showed that by inducing EKI, the flow can be developed into chaotic state and the mixing can be enhance by the chaotic flow.

In macroflows, turbulence is commonly applied to enhance mixing. However, in microflows, due to low Re, the flows used in both active and passive micromixers are either laminar or chaotic. In 2014, Wang et al. [50], for the first time, reported that turbulent-like flow in microfluidics can be realized when the flow is forced electrokinetically. In this EK forced flow, where there is an initial conductivity gradient, the viscous effect can be overcome by electrostatic force when the electric Rayleigh number is sufficiently high, although the Re in microfluidics is commonly very low. This investigation provides a new perspective for the study of EK flows and its potential future applications. In 2016, Wang et al. [51,52] characterized this micro electrokinetic (µEK) turbulence, its velocity and scalar statistics, and observed some important and typical features of turbulence which was only observed in macroflows with very high Re flows, e.g., Kolmogorov −5/3 spectrum of velocity fluctuation. Subsequently, Zhao et al. [53] evaluated the mixing effect with single-point laser-induced fluorescence (LIF) technology. A 77% and 92% mixing efficiency can be realized at 10 and 100 µm from inlet. The mixing time is 5 and 50 ms respectively. Interestingly, they found that the mixing near the bottom of the microchannel is equivalently fast as that at the centerline. Generally, due to the wall viscosity, the mixing in the center of the channel should be much stronger than that near the wall. Based on this phenomenon, they predicted that large-scale velocity structures could exist in the µEK turbulence and proposed the existence of a large-scale circulation of flow adjacent to the bottom wall because of the unbalanced electric double layer (EDL) response under AC electric field. In other words, the unmixed fluid near the wall convects to the center area, while the mixed fluid there flows into the near wall region. Nevertheless, our current understanding of the µEK turbulence is very limited, and the large-scale circulation of flow has not been supported directly by experiments.

In the present work, we study the three-dimensional (3D) flow field of the µEK turbulence based micromixer by micro particle imaging velocimetry (μPIV), with stereoscopic method [54,55]. We focused on the mean flow fields, velocity fluctuations and vorticity fields of the flows. A large-scale circulation of flow which leads to large scale uniform mixing in the entire cross section in the µEK turbulent mixer was observed. Accompanied with the extraordinarily fast mixing near the entrance, the well-mixed fluid on relatively “large scale” convects towards wall and downstream by the large-scale circulation of flow. This investigation explains why the fluid mixing in the µEK turbulent mixing can be so fast and uniform. We hope this investigation can provide a deeper understanding on the formation and evolution of the µEK turbulence.

## 2. Experiments Setup

### 2.1. Experimental System

In this investigation, a two-dimensional (2D) micro particle imaging velocity (μPIV) system was developed on the basis of inverted epi-fluorescence microscope and was used to measure the velocity fields in the µEK turbulent flow. The system is schematically shown in the Figure 1a. This system is constituted of a 532 nm dual-pulse laser (Solo III, NewWave, Fremont, CA, USA), camera (PCO Sensicam QE, Kelheim, Germany), programmable timing unit (LaVision Inc., Göttingen, Germany), 3D translation stages, objective, dichroic mirror and others optical elements, and μPIV software (Davis 7.2 from LaVision Inc., Göttingen, Germany). The laser beam passes through the dichroic mirror, and illuminates the micromixer through the objective. The reflection and transmission wavelengths of the dichroic mirror (ZT532/640rpc from Chroma, Irvine, CA, USA) are 532 nm and 580 nm respectively. The objective is a 60X with NA = 0.85. The position of the microchip is precisely controlled by the 3D translation stage (Melles Griot, Albuquerque, NM, USA). The images of fluorescent particles (Fluoro-Max Red from Thermo Fisher, Waltham, MA, USA, 1 μm diameter) are captured by the camera after passing through the dichroic mirror and a narrow-band filter (575/50 nm from Chroma, Irvine, CA, USA). The signal noise ratio can be significantly improved after the narrow-band filter.

### 2.2. Microchannel Chip

The microchip used in the present experiment is diagramed in Figure 1b. It is constructed by three layers and fabricated by a layer-by-layer process [56]. The top and bottom layers are made of an acrylic plate (1.6 mm) and a CLAREX UV transmission filter (ASTRA Products Inc., Copiague, NY, USA, 100 μm in thickness) respectively. The middle layer is a Y-type microchannel, with two inlets and one outlet. The mixing chamber is 130 μm wide (*d*), 240 μm height (*h*), 5 mm long (*l*). It has parallel top and bottom walls, and the side walls have a total 5° divergent angle. The side walls of the middle layer are consisted of gold sheets served as electrodes. The three layers are combined by double side tapes (3M, Maplewood, NJ, USA). Two streams of fluids with different electric conductivity have been injected into the two inlets by syringe pump (HARVARD Apparatus PUMP 2000, Hayward, CA, USA) directly. The flow rate of each stream is 5 μL/min. One is pure water (stream 1) and the other is PBS buffer solution (stream 2). The conductivity ratio between the two streams is 1:5000. To realize a sharp electric conductivity interface at the entrance of mixing chamber, a splitter plate with sharp trailing edge is fabricated. Both streams are uniformly mixed with fluorescent particles. In the experiment, a 100 kHz 20 V_p-p_ sinusoidal AC electric field was provided by an arbitrary function generator (Tektronix AFG3102, Beaverton, OR, USA). 

### 2.3. Velocity Fields Measured by μPIV

2D velocity fields have been measured by the μPIV system. In each measurement, 200 image pairs have been captured in a total time of 50 s. Since the depth of correlation of fluorescent particles imaged by the 60X objective could be around 30 μm, the signal noise ratio of the fluorescent particle images is still poor, because of the out-of-focus noises. Considering the aim of the research is the large-scale velocity field, to reduce the spurious velocity vectors as a result of poor signal noise ratio, interrogation window of 128 by 128 pixels has been applied with 50% overlapping during velocity calculation.

To realize 3D measurement of mean velocity fields, z-scanning method is also applied. We captured the fluorescent particle images on different z positions with interval of Δz= 20 μm. Then, 3D mean velocity field is calculated on the basis of stereoscopic method of µPIV [54,55]. Considering the fluid is incompressible,
(1)$$\nabla \cdot \vec{U}=0$$
where $\vec{U}=U\hat{x}+V\hat{y}+W\hat{z}$ is mean velocity vector with U, V and W being the local mean velocity components in $\hat{x}$, $\hat{y}$ and $\hat{z}$ directions, respectively. On each xy plane, ∂U/∂x and ∂V/∂y can be simply calculated, therefore
(2)$$\frac{\partial W}{\partial z}=-\frac{\partial U}{\partial x}-\frac{\partial V}{\partial y}$$

Applying non-slip boundary conditions for W on the bottom wall, i.e., W(z/h=−0.5)=0, W on each xy plane can be approximately calculated as
(3)$$W\left(x,y,z\right)=\frac{\partial W\left(x,y,z\right)}{\partial z}\Delta z$$

In this investigation, considering the symmetry of microchannel, we only measure the velocity fields in the lower half of the channel, i.e., −0.5≤z/h≤0.

## 3. Governing Equation of µEK Turbulence

Electrokinetic turbulence can be generated in microfluidics as a result of strong disturbance under electrostatic force (${\vec{F}}_{e}$). The flow is governed by Navier-Stokes equation with the presence of electrostatic force, as below [52]
(4)$$\frac{\partial \vec{u}}{\partial t}+\vec{u}\cdot \nabla \vec{u}=\frac{1}{\rho }\nabla p+\nu \nabla^{2}\vec{u}+\frac{1}{\rho }{\vec{F}}_{e}$$
where $\vec{u}=u\hat{x}+v\hat{y}+w\hat{z}$ is flow velocity, u, v and w are the instantaneous flow velocity components in the microchannel components in $\hat{x}$, $\hat{y}$ and $\hat{z}$ directions respectively. *ρ* is fluid density, *p* is pressure and ν is kinematic viscosity. ${\vec{F}}_{e}$ is electric volume force. Considering ${\vec{F}}_{e}$ is purely induced through electric conductivity gradient and the fluid is incompressible, we approximately have ${\vec{F}}_{e}=\rho_{f}\vec{E}$ [50], where $\vec{E}$ is electric field and $\rho_{f}=-\varepsilon \vec{E}\cdot \nabla \sigma /\sigma$ is free charge density [48]. ε and *σ* are electric permittivity and conductivity of solution, and ∇σ is the gradient of electric conductivity.

## 4. Experiments Results and Discussion

### 4.1. Distribution of Mean Velocity Field

The local mean streamwise velocity, i.e., U, is plotted in Figure 2 and compared with theoretical velocity profiles calculated by the following equation [57,58]:(5)$$\begin{array}{c} U\left(y,z\right)=\frac{48Q}{\pi^{3}wh}\left\{\sum_{n,odd}^{\infty }\frac{1}{n^{3}}\left[1-\frac{\cosh\left(\frac{n\pi y}{h}\right)}{\cosh\left(\frac{n\pi w}{2h}\right)}\right]\sin\left[\frac{n\pi }{h}\left(z+\frac{h}{2}\right)\right]\right\}\\ {\left[1-\sum_{n,odd}^{\infty }\frac{192h}{n^{5}\pi^{5}w}\tanh\left(\frac{n\pi w}{2h}\right)\right]}^{-1} \end{array}$$
where U is the local velocity of laminar flows, Q is the flow rate, −d/2≤y≤d/2, −h/2≤z≤h/2. Equation (5) is an approximation for low Reynolds number laminar flow in straight channel with rectangular cross sections.

From Figure 2, it can be seen, the measured U in divergent channel is slightly different from that in straight channel. The measured U profiles exhibit slightly sharper peaks at the center of microchannel. This is qualitatively consistent with the numerical simulation results [59] by Goli et al. who found the mean flow velocity profile in divergent channel was sharper than that in the straight channel. Besides, restricted by the poor imaging quality due to scattering of light on electrodes, the velocity cannot be accurately measured in the region y<−40 μm, where the velocity data have been abandoned in this paper. However, it can still be distinguished that U profiles are symmetric to the center of microchannel.

The symmetric mean velocity of flow field to *x*-axis without AC electric field can be more clearly observed from Figure 3a,c,e, where z/h = 0, −0.25 and −0.5 respectively. It should be noted that, all these vector fields have been normalized by the maximum velocity module of each field. Hence, even though the velocity at z/h=−0.5 is much smaller than that at z/h= 0 and −0.25, the largest velocity vector in Figure 3e still looks quite long relative to that in Figure 3a,c.

When an AC electric field of 100 kHz and 20 Vp-p is applied, the symmetry of the mean flow field is broken. As shown in Figure 3b, at z/h = 0, we can easily find the centerline of the mean flow bias towards positive y side (i.e., low σ). This trend becomes more obvious at z/h=−0.25 and −0.5, as shown in Figure 3d,f. Especially for the latter, a recirculation region with reverse flow can be observed on the high σ side. The asymmetric mean velocity fields imply that the flow has been altered by the electric volume force on the large-scale and a large-scale 3D flow may be present.

### 4.2. Velocity Fluctuation Fields

Extraordinarily fast mixing has been realized by µEK turbulent mixer in our previous investigations [50,51,52,53]. The fast mixing can be separated into two stages: one is large-scale circulation and the other is small-scale mixing. The latter has been studied before [52], while the formation of large-scale circulation has not been investigated.

In μEK turbulence, mixing of fluids relies on flow velocity fluctuations. Both $u_{rms}=\sqrt{\bar{u^{{}^{\prime }2}}}$ and $v_{rms}=\sqrt{\bar{v^{{}^{\prime }2}}}$ are plotted in Figure 4. In Figure 4a, on the z/h = 0 plane, $u_{rms}$ bias towards low σ side initially, and then becomes symmetric to *x*-axis at downstreams which is also consistent with previous investigations [5]. Similar results can be found in Figure 4c,e where z/h = −0.25 and −0.5 near the bottom. It can also be found that, the largest $u_{rms}$ does not emerge at the mid plane, but slightly towards the bottom wall, e.g., z/h = −0.25. So far the reason is not clear.

In Figure 4b, on the z/h = 0 plane, we can find $v_{rms}$ also bias towards low σ side initially, and then becomes symmetric at downstreams. The peak $v_{rms}$ at z/h = −0.25 plane (Figure 4d) is also larger than that of the other z positions (Figure 4b,f). Furthermore, since the electric conductivity gradient is parallel to the AC electric field, the electric volume force near entrance is primarily in y-direction, and accordingly $v_{rms}$ is mainly larger than $u_{rms}$ near entrance. When x/d is beyond 0.5, $u_{rms}$ and $v_{rms}$ become close to each other and the flow is more isotropic at downstream.

From Figure 4, it can be concluded the strongest velocity fluctuations happen at the low σ sides near entrance and slightly towards bottom wall. Accordingly, the mixing of the two fluids with different electric conductivity should be primarily realized at this region. However, the extraordinarily fast mixing in this local region is not sufficient to generate a uniform mixing in the entire micromixer observed previously [53]. It seems other large-scale flow is involved in to transport the mixed fluids to the rest of regions.

### 4.3. 3D Mean Velocity Field

3D mean velocity fields are plotted in Figure 5. Figure 5a shows the distribution of $\vec{U}$ of unforced flow at three different yz planes, where x/d = 0.06, 0.31, and 0.56. The top and bottom walls of the micromixer are parallel. Hence, $\vec{U}$ of unforced flow exhibits nearly parallel distributions in these yz planes, although a small W is inevitable as a result of nonuniform inlet flow and the limited accuracy of algorithm in Equation (3).

When the AC electric field is applied, a solenoid-type 3D mean flow field is rapidly developed, as shown in Figure 5b. In the yz plane of x/d = 0.06, highly negative W can be found at the low σ sides and the fluid flows towards downstream and bottom wall. In contrast, in the yz planes of x/d= 0.31 and 0.56, highly positive W can be found at the low σ sides. The flow climbs towards the top wall at downstream. The sampling period is 50 s, corresponding to four times of the large circulation time, approximately sufficient for mean velocity measurement in the ergodic process.

The solenoid-type 3D mean flow field is crucial for the extraordinarily fast and uniform mixing in the micromixer. The mixing process can be realized with two steps. First, the two fluids are highly mixed by the strong velocity fluctuations at the low σ sides near entrance, as diagramed in Figure 6a. Then, as can be seen from Figure 6b, the well-mixed fluid in the local region is transported to the rest regions of micromixer by the convection of 3D mean flow field. This is consistent to the investigation by Zhao et al. [53], which predicts the existence of large-scale circulation in the cross section.

### 4.4. x-Directional Mean Vorticity

The vorticial structures of large-scale circulation in mean flow field can be more visible from the x-directional mean vorticity ($\Omega_{x}=\frac{\partial W}{\partial y}-\frac{\partial V}{\partial z}$). As plotted in Figure 7a, near the entrance, the flush of flow towards bottom wall can be clearly found in the positive y region (i.e., low σ sides). A large region of negative $\Omega_{x}$ can be found near the center of the micromixer, and accordingly, a large-scale clockwise circulation of flow can be present. In contrast, weaker but positive $\Omega_{x}$ can be found in the cross-sectional regions at x/d= 0.31 (Figure 7b) and 0.56 (Figure 7c). The primary large-scale flow is in counter-clockwise direction.

## 5. Discussions

So far the cause of the observed large-scale solenoid-type mean flow is not fully understood. The generation of the large-scale 3D mean flow could be attributed to the unbalanced electroosmotic flows (EOFs) on both bottom and top wall of the micromixer. On the wall, electric double layers exist on the interfaces of solution and solid walls. The thickness of EDLs, estimated by Debye length $\lambda =\sqrt{\varepsilon k_{B}T/2e^{2}N_{A}c_{\infty }z_{v}^{2}}$, is inversely proportional to ion concentration $c_{\infty }$. Here, $k_{B}$ is Boltzmann constant, *T* is temperature, e is elementary charge, $N_{A}$ is Avogadro constant and $z_{v}$ is valence. Thus, on both low σ and high σ sides, λ is different. Driven by the AC electric field, a transient and oscillating EOFs can be generated adjacent to the wall. The resultant transient velocity ($v_{e}$) of EOFs of generalized Maxwell fluids in y direction can be theoretically calculated by inverse Laplacian transform as following [60]
(6)$$v_{e}\left(y,t\right)=\frac{\varepsilon E_{y}\zeta }{\mu }\frac{1}{2\pi i}\int_{c-i\infty }^{c+i\infty }\frac{K^{2}\left(1+i\tau \omega \right)}{\left(s-i\right)\left(K^{2}-\beta^{2}\right)}\left(\frac{\cosh\beta \frac{y}{h}}{\cosh\beta }-\frac{\cosh K\frac{y}{h}}{\cosh K}\right)e^{s\omega t}ds$$
where τ=ε/σ is relaxation time, $E_{y}$ is the y-directional component of electric field, ζ is the zeta potential, $\omega =2\pi f_{f}$ and $f_{f}=100\;\mathrm{kHz}$ is AC frequency, K=h/λ is a length ratio of height of the micro-channel to Debye length, $\beta^{2}=\left(1+s\tau \right)s$, s is a variable and c is a positive real constant for inverse Laplacian transform. Although Equation (6) is mainly for Maxwell fluids, it can be approximately applied for Newtonian fluids when τ is sufficiently small.

Let s=c+iq, with q being a real number, Equation (6) can also be rewritten as
(7)$$v_{e}\left(y,t\right)=\frac{\varepsilon E_{y}\zeta }{\mu }\frac{1}{2\pi i}\int_{-\infty }^{\infty }\frac{K^{2}\left(1+i\tau \omega \right)}{\left(c+iq-i\right)\left(K^{2}-\beta^{2}\right)}\left(\frac{\cosh\beta \frac{y}{h}}{\cosh\beta }-\frac{\cosh K\frac{y}{h}}{\cosh K}\right)e^{\left(c+iq\right)\omega t}dq$$
where $\beta^{2}=\left[1+\left(c+iq\right)\tau \right]\left(c+iq\right)=c+iq+{\left(c+iq\right)}^{2}\tau =\left(c+c^{2}\tau -q^{2}\tau \right)+iq\left(1+2c\tau \right)$. If we further take a temporal averaging on $v_{e}\left(y,t\right)$, the mean flow velocity of EOF driven by AC electric field becomes
(8)$$\bar{v_{e}\left(y,t\right)}=\frac{\varepsilon E_{y}\zeta }{\mu }\frac{1}{2\pi i}\int_{-\infty }^{\infty }\frac{K^{2}\left(1+i\tau \omega \right)}{\left(c+iq-i\right)\left(K^{2}-\beta^{2}\right)}\left(\frac{\cosh\beta \frac{y}{h}}{\cosh\beta }-\frac{\cosh K\frac{y}{h}}{\cosh K}\right)\bar{e^{\left(c+iq\right)\omega t}}dq$$

Considering
(9)$$\bar{e^{\left(c+iq\right)\omega t}}=\frac{\omega }{2\pi }\int_{0}^{\frac{2\pi }{\omega }}e^{\left(c+iq\right)\omega t}dt=\frac{1}{2\pi \left(c+iq\right)}\left[e^{2\pi \left(c+iq\right)}-1\right]$$
thus,
(10)$$\bar{v_{e}\left(y,t\right)}=\frac{\varepsilon E_{y}\zeta }{4\pi^{2}\mu i}\int_{-\infty }^{\infty }\frac{K^{2}\left(1+i\tau \omega \right)}{\left(c+iq-i\right)\left(K^{2}-\beta^{2}\right)\left(c+iq\right)}\left(\frac{\cosh\beta \frac{y}{h}}{\cosh\beta }-\frac{\cosh K\frac{y}{h}}{\cosh K}\right)\left[e^{2\pi \left(c+iq\right)}-1\right]dq$$

From Equation (10), it can be inferred that the mean flow velocity could be non-zero, and dependent on $E_{y}$, λ and τ which are all directly determined by electric conductivity σ. It is difficult to solve Equation (10) and obtain a formula for $\bar{v_{e}\left(y,t\right)}$ theoretically. However, evaluated from the numerical investigation of Yin et al. [60], a larger τ or smaller σ could provide higher $\bar{v_{e}\left(y,t\right)}$ qualitatively. In the meanwhile, in the low σ side, $E_{y}$ is stronger and λ is larger. We can qualitatively conclude $\bar{v_{e}\left(y,t\right)}$ could be larger in the low σ side than that in the high σ side. This can result in strong pushing along y direction near the wall and form a more complicated 3D mean flow.

In addition, ζ, which also depends on material property, could be different as well on both the top and bottom interfaces, because the top and bottom walls are consisted of CLAREX UV transmission filter and acrylic respectively. From Equation (10), it is found that different ζ can result in different transient velocity of the EOFs. This could be another reason that generates the large-scale solenoid-type mean flow.

## 6. Conclusions

In this manuscript, we studied the 3D mean and fluctuation flow field in an µEK turbulent mixer by μPIV. By applying stereoscopic method [54,55], the 3D mean flow field can be calculated. A large-scale solenoid-type 3D mean flow field has been observed. The extraordinarily fast mixing process that observed in previous investigations [50,51,53] can be attributed to two steps. First, the two fluids are highly mixed as a result of the strong velocity fluctuations near entrance, primarily at the low σ sides and bias to bottom wall. Then, the well-mixed fluid in the local region convects to the rest regions of micromixer by the convection of large-scale solenoid-type 3D mean flow field. This investigation supports the hypothesis of Zhao et al. [53] who predicts the existence of large-scale circulation in the turbulent micromixer. The mechanism of the large-scale 3D mean flow can be attributed to the coexistence of inverse energy cascade of µEK turbulence and unbalanced EOFs on the bottom wall (or top wall as well). We hope this investigation can provide a deeper understanding on the formation and evolution of µEK turbulence.

## Figures and Tables

**Figure 1 micromachines-11-00813-f001:**
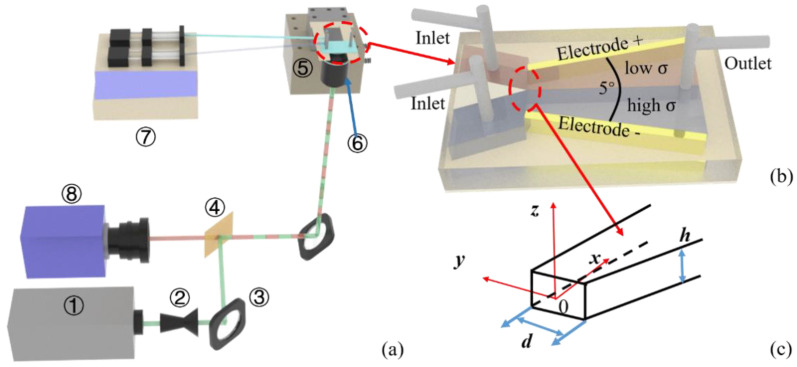
Schematic diagram of experimental system and microchannel. (**a**) Schematic diagram of experimental system, include: ① 532 nm laser, ② beam expander, ③ reflect mirror, ④ dichroic mirror, ⑤ translation stage, ⑥ objective, ⑦ syringe pump, ⑧ camera; (**b**) Schematic diagram of the µEK turbulent mixer; (**c**) Coordinate system.

**Figure 2 micromachines-11-00813-f002:**
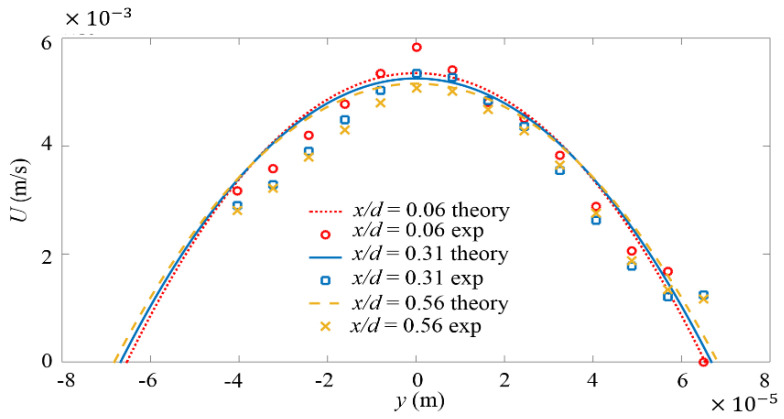
Profile of *x*-directional mean velocity U within the micromixer and at three different x positions without forcing. Dashed lines represent theoretical results. Symbols indicate experimental results from μPIV.

**Figure 3 micromachines-11-00813-f003:**
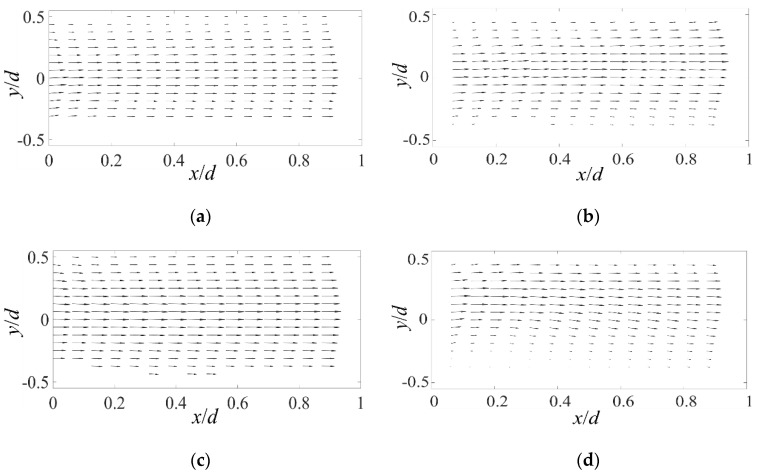
2D mean velocity fields at different z positions, with and without applying AC electric fields. The vector length has been normalized by the maximum velocity magnitude. (**a**,**c**,**e**) Unforced, and (**b**,**d**,**f**) forced. (**a**,**b**) z/h=0, (**c**,**d**) z/h = −0.25 and (**e**,**f**) z/h=−0.5.

**Figure 4 micromachines-11-00813-f004:**
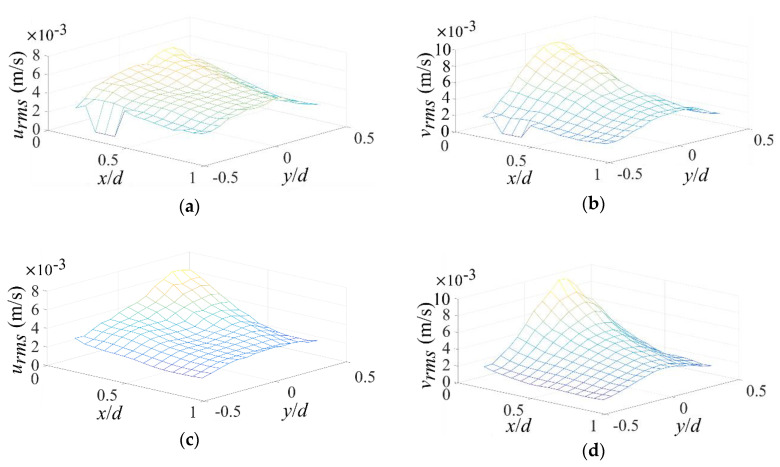
Velocity fluctuations under forcing evaluated by $u_{rms}$ and $v_{rms}$ which are the root-mean-square values of $u^{\prime }$ and $v^{\prime }$. (**a**,**c**,**e**) $u_{rms}$ at different z positions. (**b**,**d**,**f**) $v_{rms}$ at different z positions. (**a**,**b**) z/h=0. (**c**,**d**) z/h=−0.25. (**e**,**f**) z/h=−0.5.

**Figure 5 micromachines-11-00813-f005:**
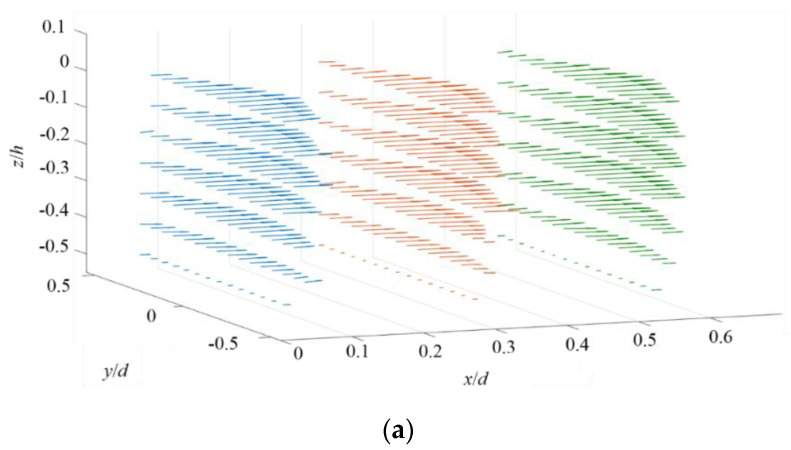
3D velocity vector on the yz planes at three different x positions (x/d = 0.06, 0.31, and 0.56). (**a**) Unforced, and (**b**) forced.

**Figure 6 micromachines-11-00813-f006:**
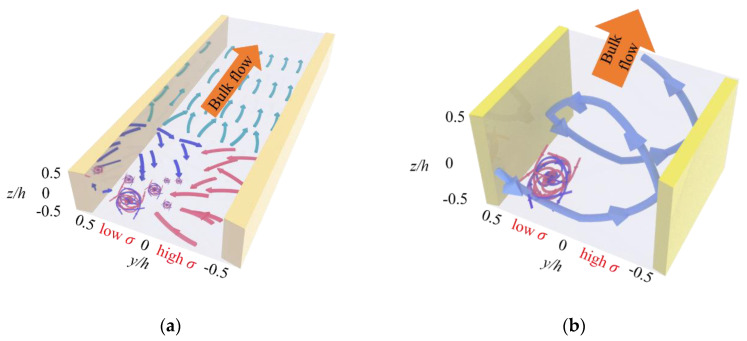
Schematic of the extraordinarily fast mixing process in the μEK turbulent mixer. (**a**) An extraordinarily fast mixing is achieved adjacent to the entrance, primarily on the low σ side and bias to negative z/h region, as indicated by the highly tangled arrows. (**b**) The well-mixed fluids that shown in (**a**) by the highly tangled arrows are convectively transported by the solenoid-type 3D mean flow (bold arrows) in the cross section and downstream. An overall uniform mixing can be rapidly realized.

**Figure 7 micromachines-11-00813-f007:**
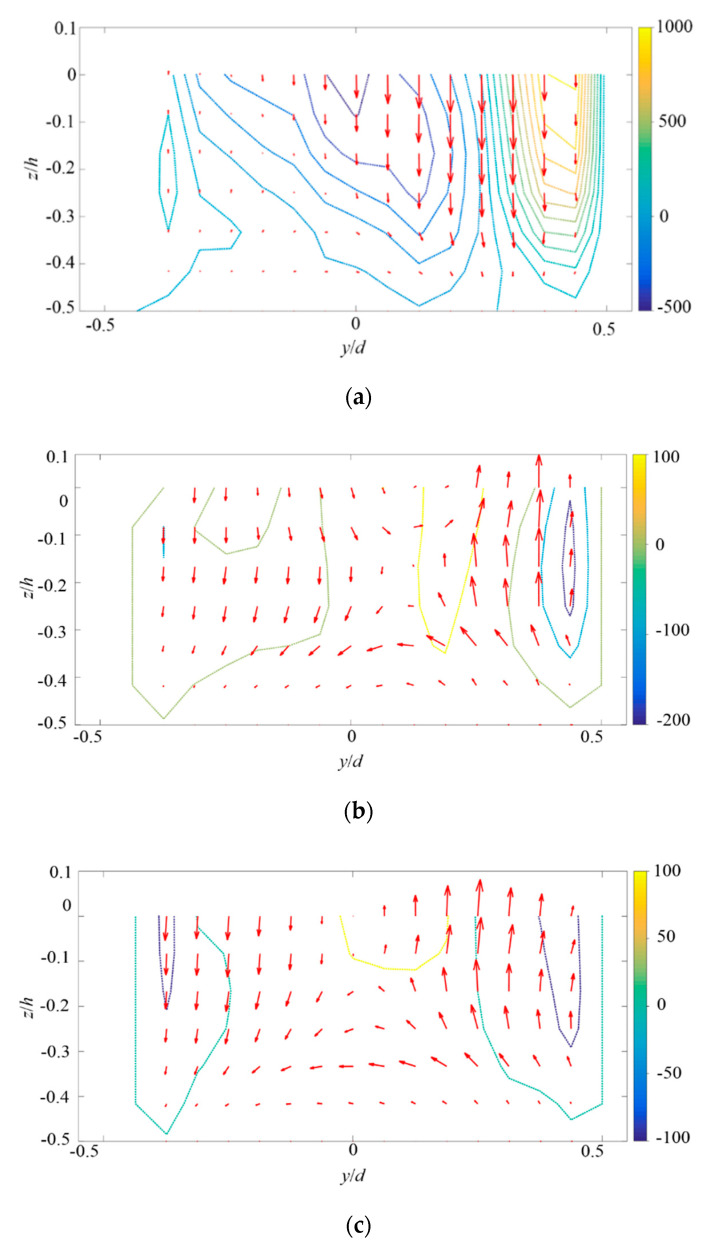
x-directional vorticity on the yz planes of three different x positions. The velocity vector in the yz planes are also plotted. (**a**) x/d = 0.06, (**b**) x/d = 0.31, and (**c**) x/d = 0.56.

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
