# Peer review of "Large-Scale Flow in Micro Electrokinetic Turbulent Mixer"

_micromachines, 2020, doi:10.3390/mi11090813_

Round 1

Reviewer 1 Report

The following comments may help the authors improving the manuscript:

  1. In the introduction, the authors need to define what electrokinetic flow is, see for example in the introduction of this ref: https://link.springer.com/content/pdf/10.1007/s10404-017-1866-y.pdf (Microfuid Nanofuid (2017) 21:37 DOI 10.1007/s10404-017-1866-y)
  2. The authors have not in the manuscript described in detail how the microfluidic chips were made. Please elaborate.
  3. What is the mixing time of this device compared to the other available, for example, the mixer made from fused silica, Micromachines20178(1), 16; https://doi.org/10.3390/mi8010016

Author Response

Please see the attached reply letter. 

Reviewer 2 Report

Dear Authors,

in your manuscript, the following points should be added / changed to further improve it:

  • References: Most of them are relatively old, there is only one from 2019, one from 2018, none from 2020. Please check the recent literature. Please correct also the formatting; there is no need for a full stop behind the journal name, but a lot of spaces missing, commas instead of semicola, etc.
  • The template is outdated, the footer shows the wrong year.
  • The affiliations 2 and 3 are mixed.
  • Abstract: What do you mean by "bias to the bottom wall"? What are EOFs?
  • line 50-51: according to your formula, the diffusion time scale is not proportional to l, but to l².
  • line 92: What is meant with "-5/3 spectrum"?
  • line 99: EDL?
  • line 138: Which shape did the alternating field have, is it sinus-like or rectangular or ...?
  • Fig. 1: Please make (a) etc. a little bit larger, in the moment they are less visible than the numbers in (a).
  • For Eq. 4 a reference should be given.
  • Generally, what is the difference between velocities given with capital and with small letters?
  • In the text, you state that you cannot measure velocities for y < -40 um. Then I would expect to see in Fig. 3 only approx. the upper half ... or symmetric halves, as also mentioned in the text. Why is both not the case here?
  •  
